# Preparation, Microstructure and Thermal Properties of Aligned Mesophase Pitch-Based Carbon Fiber Interface Materials by an Electrostatic Flocking Method

**DOI:** 10.3390/nano14050393

**Published:** 2024-02-21

**Authors:** Baoliu Li, Yudan Qin, Fang Gao, Chenyu Zhu, Changchun Shan, Jianguang Guo, Zhijun Dong, Xuanke Li

**Affiliations:** 1Hubei Province Key Laboratory of Coal Conversion and New Carbon Materials, Wuhan University of Science and Technology, Wuhan 430081, China; qinyudan@wust.edu.cn (Y.Q.); gaofang0415@wust.edu.cn (F.G.); zhuchenyu@wust.edu.cn (C.Z.); guojianguang@wust.edu.cn (J.G.); dongzj72@sohu.com (Z.D.); 2Hubei Province Pilot Base on Coal Conversion and New Carbon Materials, Wuhan University of Science and Technology, Wuhan 430081, China; 3Baowu Carbon Technology Co., Ltd., Shanghai 201999, China; shanchangchun@baosteel.com

**Keywords:** mesophase pitch-based carbon fibers, carbon nanotubes, vertical array, interface materials, thermal conductivity

## Abstract

The mesophase pitch-based carbon fiber interface material (TIM) with a vertical array was prepared by using mesophase pitch-based short-cut fibers (MPCFs) and 3016 epoxy resin as raw materials and carbon nanotubes (CNTs) as additives through electrostatic flocking and resin pouring molding process. The microstructure and thermal properties of the interface were analyzed by using a scanning electron microscope (SEM), laser thermal conductivity and thermal infrared imaging methods. The results indicate that the plate spacing and fusing voltage have a significant impact on the orientation of the arrays formed by mesophase pitch-based carbon fibers. While the orientation of the carbon fiber array has a minimal impact on the shore hardness of TIM, it does have a direct influence on its thermal conductivity. At a flocking voltage of 20 kV and plate spacing of 12 cm, the interface material exhibited an optimal thermal conductivity of 24.47 W/(m·K), shore hardness of 42 A and carbon fiber filling rate of 6.30 wt%. By incorporating 2% carbon nanotubes (CNTs) into the epoxy matrix, the interface material achieves a thermal conductivity of 28.97 W/(m·K) at a flocking voltage of 30 kV and plate spacing of 10 cm. This represents a 52.1% increase in thermal conductivity compared to the material without TIM. The material achieves temperature uniformity within 10 s at the same heat source temperatures, which indicates a good application prospect in IC packaging and electronic heat dissipation.

## 1. Introduction

The integration of microelectronic devices has witnessed a notable rise, resulting in a simultaneous reduction in space and increase in power. Consequently, this has led to a significant surge in the generation of localized heat. To address the issue of heat conduction between the radiator and the heat source [1], it is necessary to install interface material (TIM) with high thermal conductivity and high elasticity between the radiator and the heat source gap [2,3] to realize the rapid conduction of heat from the heat source to the radiator [4,5], reducing its working temperature. Furthermore, it will enhance the operational stability, dependability, and lifespan of electronic components. Currently, thermal paste, thermal conductive silica gel, thermal conductive tape, phase change materials, thermal gaskets, and others comprise the majority of commercial interface materials. The thermal conductive gasket’s superior thermal conductivity, compressive performance, and reusable nature facilitates disassembly which has garnered considerable interest from researchers [6,7,8,9].

In order to improve the thermal conductivity of the thermal gasket, the most common method is adding high thermal conductivity filler to the flexible polymer matrix [10,11,12,13,14,15], such as graphite nanosheet [13], boron nitrided [14], aluminum nitrided [15], single/multi-wall carbon nanotubes [16,17,18], graphene [19,20,21,22], carbon fiber [23,24,25,26], etc. Thermal fillers frequently employ mesophase pitch-based carbon fibers owing to their advantageous characteristics, including a controllable aspect ratio design, effective electromagnetic shielding, and high axial thermal conductivity (~1100 W/(m∙K)). Zifeng Yu et al. [27] used the method of electrostatic flocking to make the mesophase pitch-based carbon fibers arranged in an orientation in epoxy resin. When the filling amount of carbon fibers was 13.4 wt%, the thermal conductivity of the interface material was 15.3 W/(m·K). Kojiro Uetani et al. [28] prepared a carbon fiber/fluorine rubber interface material by combining a vertically arranged mesophase pitch-based carbon fiber skeleton with fluorine rubber using electrostatic flocking technology. When the carbon fiber content was 13.2 vol%, the thermal conductivity of the composite material in the thickness direction was 23.3 W/(m∙K). Tengxiao Ji et al. [29] modified CNT on mesophase pitch-based carbon fibers via chemical deposition and oriented the carbon fibers via electrostatic flocking in order to produce a carbon fiber-carbon nanotube/silicone rubber composite material. At a carbon fiber-carbon nanotube content of 1.5% by weight, the composite exhibited a thermal conductivity of 6.57 W/(m·K). 

Utilizing an electric or magnetic field, the aforementioned study produces high-orientation carbon fiber interface material. Based on the above research work, this paper prepared carbon fiber interface material using self-made mesophase pitch-based short-cut fibers and 3016 epoxy resin as primary materials, and carbon nanotubes as additives by electrostatic flocking and resin pouring processes. The effects of flocking voltage and plate spacing on the orientation of carbon fiber array and the thermal conductivity and mechanical properties of the interface materials were systematically investigated. We analyzed the effects of varying carbon nanotube addition applied to the resin matrix on the thermal behavior and mechanical properties of the interface materials. 

## 2. Experiment

### 2.1. Preparation of Vertical Array Carbon Fiber (VACF)

Figure 1 shows the properties of mesophase pitch-based short-cut fibers. Figure 1a,b show that the diameter of carbon fibers was mainly concentrated between 14 and 24 μm, and the length of carbon fibers was mainly concentrated between 0.8 and 1.8 mm. Figure 1c shows the typical X-ray diffraction (XRD) patterns from the carbon fiber powder. The intensities of the (002) and (110) diffraction peaks of the powder were very strong, which indicated that mesophase pitch-based short-cut fibers had a high graphitization degree and complete graphite crystal structure. As depicted in Figure 1d, the cross section of the mesophase pitch-based short-cut fibers shows a large angle split radial structure and clearly visible graphite microchip layer. 

The above short-cut fibers were used as raw materials, a blade coater was utilized to uniformly apply silicon rubber to the polyimide film, which had a thickness of 200 μm. The silicone rubber-coated polyimide film was affixed to the upper plate of the electrostatic flocking apparatus, while the 1 mm long mesophase pitch-based short-cut fibers were uniformly distributed on the lower plate. The plate spacings were 8 cm, 10 cm, and 12 cm, and the flocking voltages were 20 kV, 30 kV, and 40 kV, respectively. By applying electric fields of varying intensities, the carbon fibers were oriented in a specific direction on the polyimide film. Following flocking, the sample was subjected to an electric heating constant temperature air drying furnace at 60 °C for four hours in order to cure the silicone rubber in preparation for the carbon fiber array (VACF). By computing the orientation angle of the carbon fibers, the proportion of mesophase pitch-based carbon fibers with an angle greater than 45° to the bottom surface in relation to the total number of carbon fiber arrays was denoted as “P”.

### 2.2. Preparation of VACF/Epoxy Interface Material

Following the preparation of glue A and glue B from 3016 epoxy resin in a mass ratio of 1:1, carbon nanotubes of varying qualities were introduced and thoroughly combined. Following a vacuum standing period of 20 min, the bubbles were eliminated. After pouring the mixed epoxy resin solution onto the VACF, a 200 μm silicone rubber cover plate was utilized to envelop the surface. Appropriate weight was applied to the cover surface, and the thickness of the sample was controlled by limiting the place. The upper and lower silicone rubber of the cover plate could be removed after standing and curing for 24 h at room temperature to obtain VACF/epoxy interface material. To enhance the thermal conductivity of the interface material, various mass fractions of CNTs were incorporated into a 3016 epoxy resin matrix, and one of the VACFs was selected as the raw material. The process flow diagram of VACF/epoxy-mCNT interface materials is depicted in Figure 2. The samples were named VACF/epoxy-mCNTs according to the amount of CNTs added, where the m values were 0, 0.5, 1.0, 1.5, and 2.0. The filling rate of fibers in VACF/epoxy-mCNTs was about 6.30 wt% under different electrostatic fusing processes, which is lower than the value reported in the above research work [27,28]. The microstructure and thermal properties of the interface were analyzed as shown below.

### 2.3. Detection and Analysis

The microstructure and morphology of VACF/epoxy-mCNT interface materials in different directions were observed using a JSM-7601F field emission scanning electron microscope (SEM) (JSM-7601F, JEOL, Tokyo, Japan). The cross section hardness of the interface material was tested using a shore hardness tester LX-A, and five data points were measured for each sample and averaged. The surface XRD of the VACF/epoxy-mCNT interface materials were subjected to X-ray diffraction (XRD, Philip X’Pert MPD Pro, PANalytical, Almelo, The Netherlands) utilizing Cu Kα radiation (λ = 0.15406 nm) at an accelerating voltage and current of 40 kV and 30 mA, respectively.

The cross section thermal conductivity of VACF/epoxy-mCNT interface materials was assessed using an indirect method. The specimen was segmented into blocks measuring 10 mm × 10 mm × (0.6–1) mm, and the thermal diffusivity of the material was ascertained using a laser-flash diffusivity apparatus (LFA 457, NETZSCH, Selb, Germany) under ambient temperature conditions. The thermal conductivity of the material was computed by applying the formula λ = α × ρ × Cp, where α represents the thermal diffusivity, ρ denotes the volume density, and Cp signifies the specific heat capacity. The heat transfer abilities of the interface material were determined using infrared thermography from the FOTRIC 326L (HUEIKO, Wuxi, China). The VACF/epoxy-mCNT interface material was positioned on a heated table maintained at a constant temperature, and the temporal variations in surface temperature of the interface material were documented.

## 3. Results and Discussion

### 3.1. Morphology and Structure of VACF

Figure 3 shows the SEM and orientation distribution histogram of the carbon fiber array after electrostatic flocking. Figure 3(a_1_–a_3_) shows that the orientation of the carbon fiber array is obviously different under different flocking processes. When the plate spacing was 10 cm, the flocking voltage of 30 kV and 40 kV were less oriented than the flocking voltage of 20 kV. As depicted in Figure 3(b_1_–b_3_), when the flocking voltage was 20 kV, the array orientation of the carbon fiber improved as the plate spacing increased. This was primarily due to the high flocking voltage, the increase in the electrostatic attraction between the plates, and the increase in the speed of the carbon fiber movement between the plates. As a consequence, the carbon fibers were unable to promptly adjust their orientation from the horizontal to the vertical direction. In addition, the excessive moving speed of carbon fiber between plates made it easy to impact the fiber array on the upper plate, resulting in the destruction of the already oriented carbon fiber array, while the electrostatic flocking process could make the carbon fiber powder realize the vertical array orientation. 

Figure 3c,d show that the *p* values of the carbon fiber array were 0.91, 0.91, and 0.89, respectively, when the distance between plates was 10 cm and the flocking voltages were 20 kV, 30 kV, and 40 kV. At a flocking voltage of 20 kV and plate spacings of 8 cm, 10 cm, and 12 cm, the *p* values of the carbon fiber array were 0.91, 0.91, and 0.92, respectively. These findings suggested that the voltage applied during flocking and the distance between the plates significantly affected the alignment of carbon fibers in the array. The optimal orientation of the carbon fiber array was achieved when the flocking voltage was set at 20 kV and the plate spacing was maintained at 12 cm.

### 3.2. The Cross Section Morphology and Structure of VACF-0CNT/Epoxy Interface Material 

As depicted in Figure 4, the SEM images and surface XRD of the VACF/epoxy-0CNT interface material illustrate its cross section morphology and structure. The alignment of carbon fibers in the VACF/epoxy-0CNT interface material was predominantly at an angle ranging from 70° to 90°, as illustrated in the figure. With the increase in flocking voltage, the consistency of the carbon fiber inclination angle of the cross section became worse when the plate spacing was 10 cm. When there was a flocking voltage of 20 kV and plate spacing of 10 cm, the carbon fibers in the VACF/epoxy-0CNT interface material were mainly arranged in a vertical array at an angle of 90°. The arrangement of the carbon fibers into an inclination angle array was predominant when the flocking voltage was set to 40 kV. Conversely, a minority of the carbon fibers were arranged in a vertical array at an angle of 90°. This observation aligns with the hypothesis that the electrostatic flocking process influences the orientation of the carbon fiber array. Figure 4c,d show the surface XRD curve of the VACF/epoxy-0CNT interface material under different flocking processes. The intensities of the (002) and (110) diffraction peaks of the graphite characteristic peak were weak due to the high resin content.

### 3.3. Thermal Conductivity, Mechanical Properties and Heat Transfer Abilities of VACF-0CNT/Epoxy Interface Material

Table 1 shows the thermal conductivity and shore hardness of VACF/epoxy-0CNT interface material under different molding processes. The table presented data indicating that the thermal conductivity of the interface material increased as the plate spacing increased, while the flocking voltage remained constant. Conversely, when the plate spacing remained constant, the thermal conductivity of the interface material decreased as the flocking voltage increased. This pertained primarily to the carbon fiber array’s orientation within the VACF/epoxy-0CNT interface material. The carbon fiber array exhibited an enhanced alignment effect when accorded an adequate amount of time to align between the plates. When the flocking voltage was 20 kV and the plate spacing was 12 cm, the thermal conductivity of the interface material was the best, reaching 24.47 W/m·K. However, the thermal conductivity reported in the reference was 23.3 W/(m∙K), and the carbon fiber content was 13.2 vol% [28], which indicates that the orientation of the fiber array was better than the reference’s research work. The shore hardness of the VACF/epoxy-0CNT interface material had little relationship with the carbon fiber array orientation, which was between 41 and 46 A. This indicates that the orientation of the fiber array affects the thermal conductivity of the interface material but has little effect on the hardness of the material.

Figure 5 shows the infrared thermography of the VACF-0CNT/epoxy interface material. The VACF-0CNT/epoxy interface material was placed on a constant temperature platform and the surface temperature of the interface material at different time points was monitored by an infrared thermal imager. When the plate spacing was 10 cm, the flocking voltage of 20 kV had the fastest heating rate from room temperature up to 60 °C and a temperature uniformity of about 15 s. When the flocking voltage was 20 kV, with the increase in the plate spacing, the time for the interface material to reach temperature uniformity was shorter. 

### 3.4. The Cross Section Morphology and Structure of VACF/Epoxy-mCNT Interface Material

To enhance the thermal conductivity of the VACF/epoxy-mCNT interface material, various mass fractions of CNTs were incorporated into a 3016 epoxy resin matrix. In order to fabricate the VACF/epoxy-mCNT interfacial material, a carbon fiber array was utilized, accompanied by a flocking voltage of 30 kV and plate spacing of 10 cm. Figure 6a–e illustrate the cross sectional microstructure of an interface material consisting of VACF, epoxy, and varying quantities of CNT. The absence of evident bending and aggregation in the carbon fiber arrays, along with their uniform dispersion and vertical orientation with inclination in the cross section of the VACF/epoxy-mCNT interface material, suggests that the incorporation of CNTs will not disrupt the orientation of the carbon fiber arrays. The interface material exhibited a limited quantity of patches on its cross section subsequent to the incorporation of CNTs. As the quantity of addition increased, there was a corresponding progressive enlargement of the spots observed on the cross section of the interface material. The primary cause of the blotches was the aggregation of CNTs that occurred during dispersion. As shown in the diagram, Figure 6f depicts surface SEM image and XRD of the VACF/epoxy-mCNT interface material. The figure illustrates how the carbon fiber array traverses the interface material, with the fiber cross section being conspicuously apparent on the material’s surface and a minor quantity of carbon fiber remaining on the material’s surface. Additional evidence suggested that the carbon fiber array remained undamaged throughout the resin composite process. The intensities of the (002) and (110) diffraction peaks of the graphite characteristic peak were also very weak due to the high resin content.

### 3.5. Thermal Conductivity and Mechanical Properties of VACF/Epoxy-mCNTs Interface Materials

Figure 7 shows the curve of the relationship between the hardness of the VACF/epoxy-mCNT interface material and the concentration of CNTs. The volume fraction of carbon fibers in the VACF/epoxy-0CNTs cross section material was 6.30%, and the hardness was 48 A without added CNTs. The hardness of the VACF/epoxy-mCNT interface material increased marginally but progressively as the CNTs concentration increased. This was primarily due to the fact that the material’s hardness was highly dependent on its composition. The primary determinant of the hardness of the VACF/epoxy-mCNT interface material mainly came from the carbon fibers. The fiber volume fraction of VACF/epoxy-mCNT interfacial materials containing varying quantities of CNTs was essentially the same. However, the interface material’s hardness increased marginally with the addition of CNTs. The shore hardness of the material was 52 A when the amount of CNTs added was 2 wt%.

Figure 8 shows the curve of the thermal conductivity of the VACF/epoxy-mCNT interface material versus the amount of CNTs added. The figure illustrates that the VACF/epoxy-mCNT interface material, which was produced in the absence of carbon nanotubes, possesses a thermal conductivity of 19.05 W/(m·K). This value was determined using a fusing voltage of 30 kV and a plate spacing of 10 cm. After the addition of CNTs, the thermal conductivity of the VACF/epoxy-mCNT interface material increased markedly. When the addition amount was 0.5 wt%, the thermal conductivity of VACF/epoxy-0.5CNT interface material reached 27.11 W/(m·K). With the further increase in the amount of CNTs, the thermal conductivity of the interface material increased slowly. The thermal conductivity of VACF/epoxy-2.0CNTs interface material was 28.97 W/(m·K) when the CNT addition amount was 2.0 wt%, which was higher than the 0.5 wt% addition. As is well known, CNTs have good thermal conductivity, and the addition of CNTs can improve the interface thermal resistance between the carbon fiber array and the epoxy resin. When the CNT content reached 0.5 wt%, a thermal conductive network formed at the interface of the material, and the thermal conductivity of the interface materials increased significantly. However, as the thermal conductivity of the material continued to rise, its growth became more gradual. A specification of 0.5 wt% of the CNT content caused the material to surpass the seepage threshold.

### 3.6. Heat Transfer Abilities and Conduction Mechanism of VACF/Epoxy-mCNT Interface Material

Figure 9 shows the infrared thermography and model diagram of the VACF/epoxy-mCNT interface material. The VACF/epoxy-mCNT interface material was placed on a constant temperature platform, and the surface temperature of the interface material at different time points was monitored by an infrared thermal imager. The surface of VACF/epoxy-0CNTs had the slowest heating rate from room temperature up to 60 °C and a temperature uniformity of about 25 s. The introduction of carbon nanotubes resulted in a consistent upsurge in the heating rate of the interface material. At a carbon nanotube addition of 2.0 wt%, the VACF/epoxy-2.0CNTs achieved temperature uniformity within a span of 10 s. The addition of CNTs to the resin matrix can enhance the thermal performance of the interface material, increase the efficacy of heat conduction, and reduce the time required for the interface material to reach equilibrium temperature. As depicted in the model diagram of the VACF/epoxy-mCNT interface material, when the amount of CNT added reached a certain point, the interface material filled with CNTs had more heat conduction channels.

## 4. Conclusions

The VACF/epoxy-mCNT interface material with a vertical array was prepared by using mesophase pitch-based short-cut fibers and 3016 epoxy resin as raw materials, and carbon nanotubes as an additive. The electrostatic flocking procedure significantly influenced the carbon fiber array’s orientation. The interface material’s thermal conductivity improved as the orientation of the carbon fiber array increased. However, the shore hardness of the material was nominally unaffected by the electrostatic flocking process. The inclusion of CNTs resulted in a marginal increase in the thermal conductivity of the interface material compared to a 0.5 wt% addition, suggesting that the material reached the percolation threshold for thermal conductivity with this addition. When the flocking voltage was 30 kV, plate spacing was 10 cm, carbon nanotube addition was 2.0 wt%, the interface of thermal conductivity reached at 28.97 W/(m·K), and the same heat source temperatures realized temperature uniformity within 10 s, which has huge application prospects in IC packaging and electronic heat dissipation [7,8].

## Figures and Tables

**Figure 1 nanomaterials-14-00393-f001:**
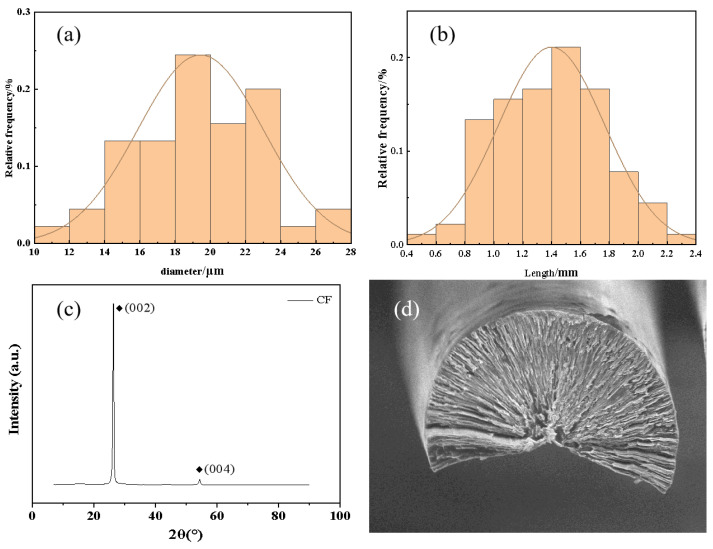
The properties of mesophase pitch-based short-cut fibers. (**a**) Diameter distribution of carbon fibers; (**b**) length distribution of carbon fibers; (**c**) XRD curve of carbon fibers; (**d**) SEM image of carbon fibers.

**Figure 2 nanomaterials-14-00393-f002:**
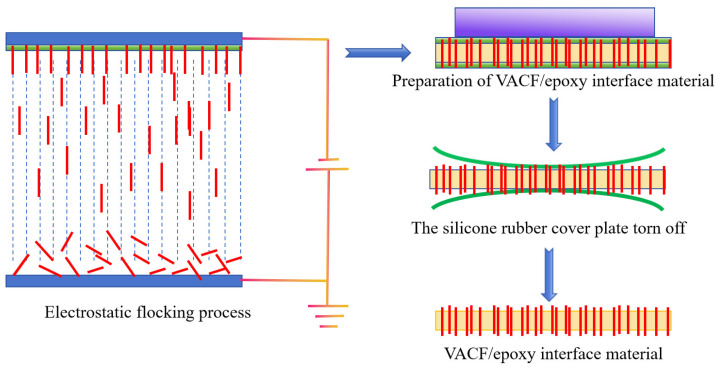
The process flow diagram of VACF/epoxy-mCNT interface materials.

**Figure 3 nanomaterials-14-00393-f003:**
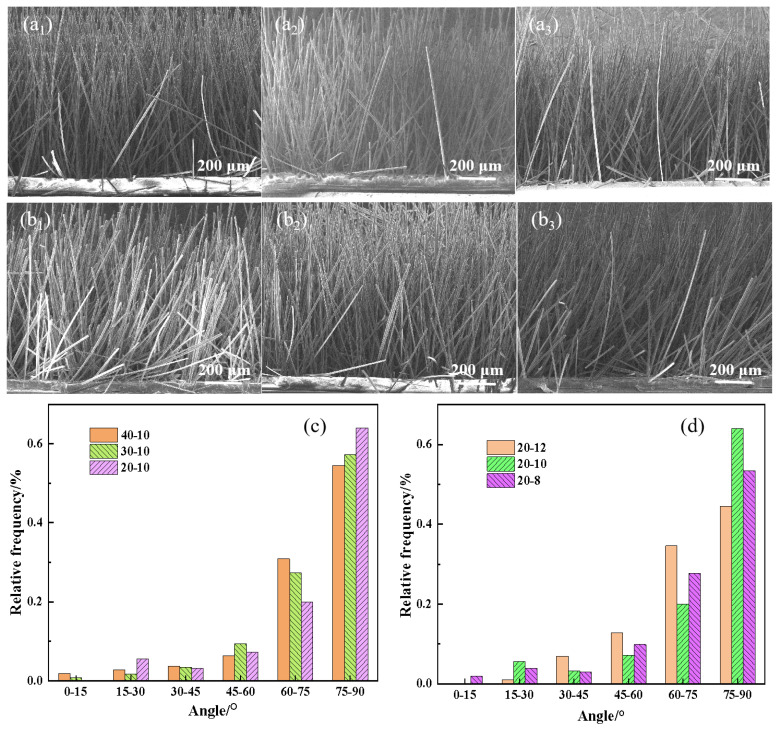
Electrostatic flocking after the carbon fiber array SEM images and orientation distribution histogram. (**a_1_**–**a_3_**) Plate spacing was 10 cm, and flocking voltages were 20 kV, 30 kV, and 40 kV. (**b_1_**–**b_3_**) Flocking voltage was 20 kV, plate spacings were 8 cm, 10 cm, 12 cm. (**c**,**d**) Orientation distribution histogram of carbon fiber array under different flocking process conditions.

**Figure 4 nanomaterials-14-00393-f004:**
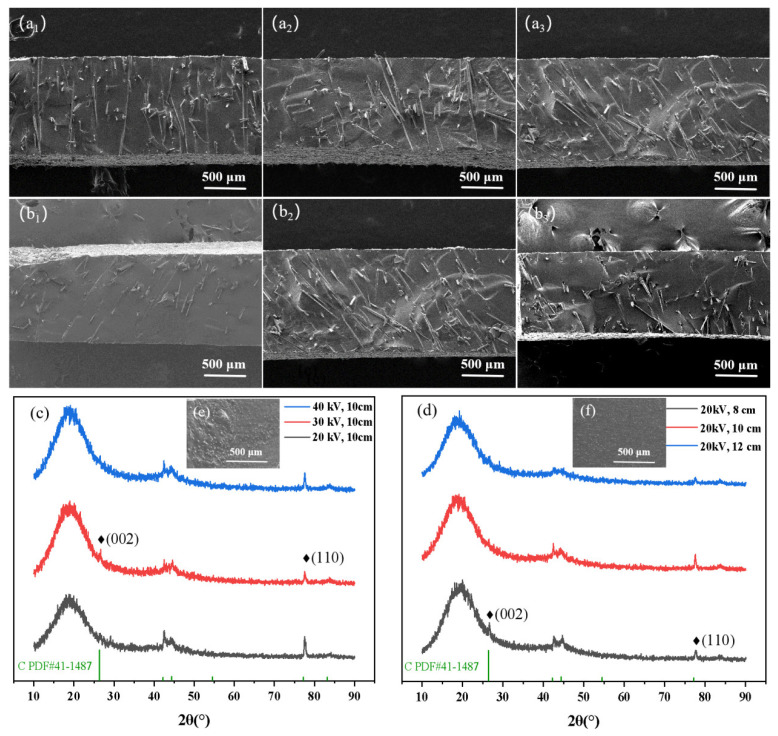
The cross section SEM images and surface XRD of the VACF/epoxy-0CNT interface material. (**a_1_**–**a_3_**) The plate spacing was 10 cm, and the flocking voltages were 20 kV, 30 kV, and 40 kV. (**b_1_**–**b_3_**) The flocking voltage was 20 kV, and the plate spacings were 12 cm, 10 cm, and 8 cm. (**c**,**d**) The surface XRD curve of the VACF/epoxy-0CNT interface material under different flocking processes. (**e**,**f**) The surface SEM images of the VACF/epoxy-0CNT interface material.

**Figure 5 nanomaterials-14-00393-f005:**
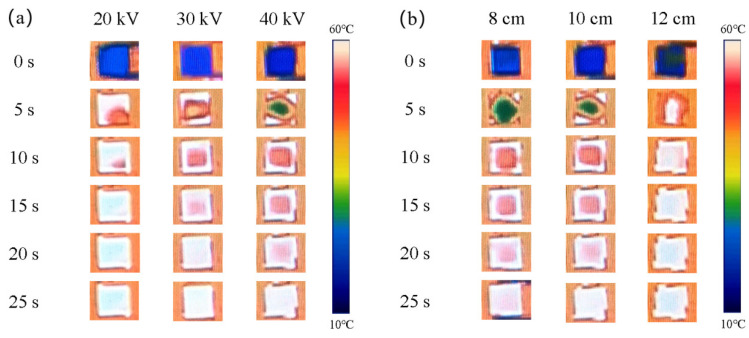
Infrared thermography of VACF-0CNT/epoxy interface material. (**a**) Plate spacing was 10 cm, and flocking voltages were 20 kV, 30 kV, and 40 kV. (**b**) Flocking voltage was 20 kV, and plate spacings were 8 cm, 10 cm, and 12 cm.

**Figure 6 nanomaterials-14-00393-f006:**
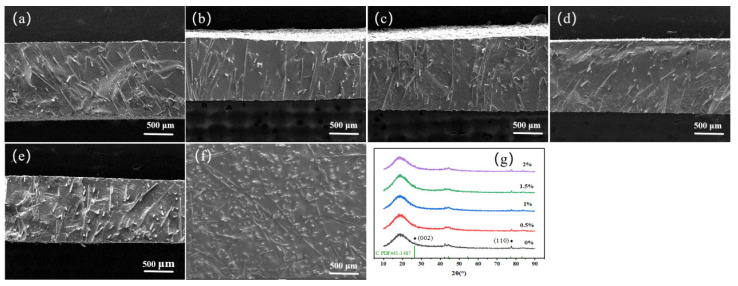
SEM images and surface XRD of the VACF/epoxy-mCNT interface material. (**a**–**e**) Cross section morphology of the interface material with 0 wt%, 0.5 wt%, 1.0 wt%, 1.5 wt%, and 2.0 wt% CNT addition, respectively. (**f**) Surface SEM image of the VACF/epoxy-mCNT interface material. (**g**) Surface XRD of the VACF/epoxy-mCNT interface material.

**Figure 7 nanomaterials-14-00393-f007:**
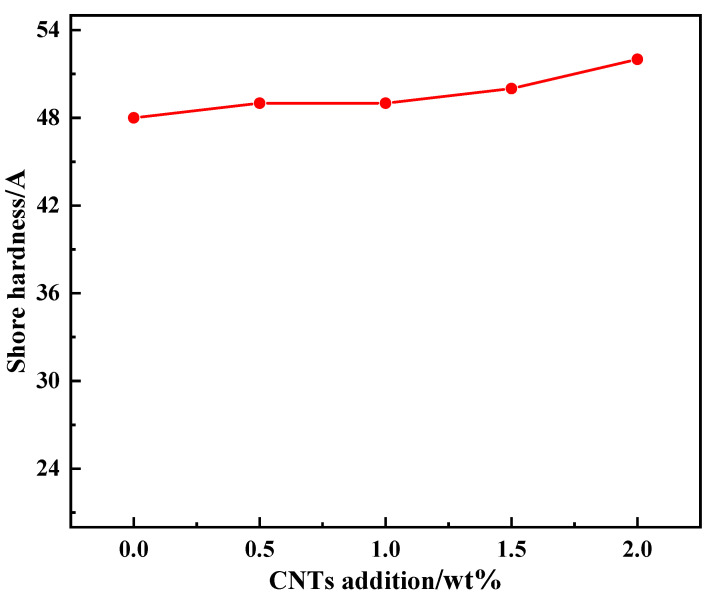
The relationship between shore hardness of VACF/epoxy-mCNT interface material and the amount of CNTs added.

**Figure 8 nanomaterials-14-00393-f008:**
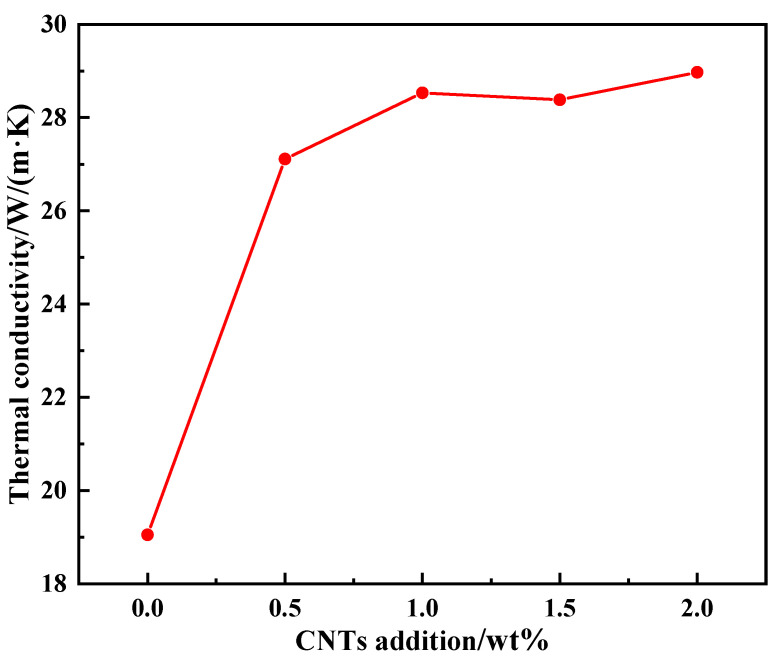
The relationship between thermal conductivity of VACF/epoxy-mCNT interface material and the amount of CNTs added.

**Figure 9 nanomaterials-14-00393-f009:**
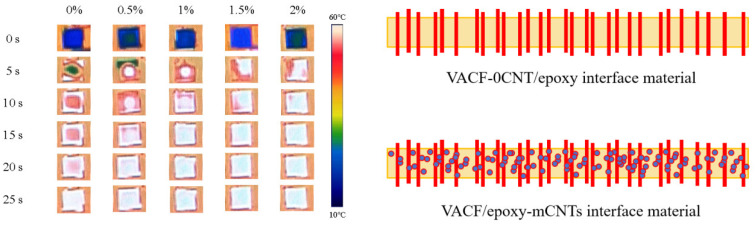
Infrared thermography and model diagram of VACF/epoxy-mCNT interface material.

**Table 1 nanomaterials-14-00393-t001:** The thermal conductivity and shore hardness of VACF/epoxy-0CNT interface material under different molding processes.

Molding Processes	20 kV/8 cm	20 kV/10 cm	20 kV/12 cm	30 kV/10 cm	40 kV/10 cm
Thermal Conductivity W/(m∙K)	12.54	16.04	24.47	19.05	13.48
Shore Hardness/A	45	41	46	46	43

## Data Availability

Data are contained within the article.

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
