# Peer review of "Preparation, Microstructure and Thermal Properties of Aligned Mesophase Pitch-Based Carbon Fiber Interface Materials by an Electrostatic Flocking Method"

_nanomaterials, 2024, doi:10.3390/nano14050393_

Round 1

Reviewer 1 Report

Comments and Suggestions for Authors

In my opinion, a very interesting article. The work describes the technology of producing thermal bonding materials from carbon fibers based on mesophase pitch, using the electrostatic flocking method. Research in this area is extremely valuable and necessary. At the same time, I have a few comments regarding the publication:

1. In point 2.1. Preparation of vertical array carbon fiber (VACF), the authors should better describe (more precisely) the method of sample preparation - number of samples, test plan, etc.

2. Section 2.3. Detection and analysis should be expanded (no photos or detailed description)

3. Figures 7 and 8 - why was linear interpolation used between measurement points?

4. In my opinion, the work lacks experiment planning, determining the number of samples, eliminating gross errors, etc.

5. In the case of analysis of the technological process, is it possible to optimize it? Are the authors able to optimize the manufacturing process? If so, it is worth presenting the results in the article.

The work is very interesting and requires minor additions.

Author Response

Dear reviewer,

Thank you very much for your comments and professional advice. These opinions help to improve academic rigor of our article. Based on your suggestion and request, we have made corrected modifications on the revised manuscript. Meanwhile, the manuscript had be reviewed and edited by language of MDPI. We hope that our work can be improved again. Furthermore, we would like to show the details as follows:

  1. In point 2.1. Preparation of vertical array carbon fiber (VACF), the authors should better describe (more precisely) the method of sample preparation - number of samples, test plan, etc.

The author’s answer: The preparation process and test plan of the sample are added in the paper. (The red part in 2.1 Preparation of vertical array carbon fiber (VACF) and 2.2 Preparation of VACF/epoxy interface material)

  1. Section 2.3. Detection and analysis should be expanded (no photos or detailed description)

The author’s answer: The detection and analysis was further modified and the test method of XRD was added. (The red part in Section 2.3. Detection and analysis).

  1. Figures 7 and 8 - why was linear interpolation used between measurement points?

The author’s answer:  The purpose of using the linear interpolation between measurement points is to show the relationship between the amount of CNTs addition and hardness and thermal conductivity more intuitively and accurately. The other graphs can lead to spurious data results.

  1. In my opinion, the work lacks experiment planning, determining the number of samples, eliminating gross errors, etc.

The author’s answer:   The experiment planning and experimental conditions are added in the paper. The test results in the paper are the average values of multiple samples, which are representative and able to eliminating gross errors. (The red part in 2.1 Preparation of vertical array carbon fiber (VACF) and 2.2 Preparation of VACF/epoxy interface material)

  1. In the case of analysis of the technological process, is it possible to optimize it? Are the authors able to optimize the manufacturing process? If so, it is worth presenting the results in the article.

The authors answer:   The technological process is basically the optimal manufacturing process. Before that, the authors did a lot of experiments and prepared a lot of discarded samples, the pictures were shown as below.

Thank you very much for your attention and time. Look forward to hearing from you.

Yours sincerely,

Baoliu Li

24 January, 2024

Hubei Province Key Laboratory of Coal Conversion and New Carbon Materials,

Wuhan University of Science and Technology,

Wuhan 430081, China

Tel:86-27-68862591

E-mail: libaoliu@wust.edu.cn

Reviewer 2 Report

Comments and Suggestions for Authors

The manuscript focuses on a topic of great interest to the reader.

In fact, mesophase pitch-based carbon fibers are most attractive candidates for thermally conductive materials and heat dissipation materials due to their low electrical resistivity and high thermal conductivity. Besides, mesophase pitch-based carbon fiber (MPCF) and its composites are considered to be promising structural and functional materials in applications such as aerospace, electronics, advanced manufacturing, and other sophisticated industry due to their high specific Young’s modulus and high thermal conductivity.

While MPCFs have excellent properties resulting from a high degree of preferred orientation of the graphite layers parallel to the fiber axis, however, due to the weak interlayer interaction between highly aligned large planar molecules, the open wedge cracking is often observed in MPCFs with radial-type ordered transverse structures, resulting in low strain at failure and low utilization rate in composite manufacturing, which severely hinders its practical application. Therefore, all research aimed at finding solutions for real practical application is held in high regard.

My comments that authors must appropriately address for the manuscript to be considered for publication are set out below:

1) what are the advances in terms of results and performance compared to the current literature on the topic covered by the authors?

2) it is necessary to explain both in the abstract, in the introduction and in the conclusions the objective of the research that led the authors to conceive this work

3) based on the results obtained, what applications are foreseen for the developed material?

4) english must be revised both in syntax and grammar

5) when describing the figures, the number of the figure should be put in front of the letter: for example, Fig.1 (a,b) instead of Fig. (a,b)

6) some characterization techniques (for example the XRD technique) were not described in the manuscript while sufficient details were not provided for others

7) how were the mechanical properties determined? please provide a description

8) the XRD image inserted in figure 6f is absolutely not readable

9) on the surface SEM images of the VACF/epoxy-0CNT interface material shown in Figure 4 (e,f), the scale bar which is absent must be inserted

10) the most up-to-date references on the topic covered must be included in the manuscript; in this regard, I suggest consulting the following articles:

·        https://doi.org/10.1007/s10853-021-06770-9

·        doi: 10.1126/sciadv.abn1905

Comments on the Quality of English Language

Moderate editing of English language required

Author Response

Dear reviewer,

Thank you very much for your comments and professional advice. These opinions help to improve academic rigor of our article. Based on your suggestion and request, we have made corrected modifications on the revised manuscript. Meanwhile, the manuscript had be reviewed and edited by language of MDPI. We hope that our work can be improved again. Furthermore, we would like to show the details as follows:

1) what are the advances in terms of results and performance compared to the current literature on the topic covered by the authors?

The author’s answer: The interface material exhibited an optimal thermal conductivity of 24.47 W/(m·K) and carbon fiber filling rate of 6.30 wt%. The interface material with higher thermal conductivity can be prepared with less fiber filling, which indicates that the fiber array orientation is better than that reported in other paper.

2) it is necessary to explain both in the abstract, in the introduction and in the conclusions the objective of the research that led the authors to conceive this work

The author’s answer: We have made a targeted modification to this problem. (The red part in the abstract, introduction and conclusions).

3) based on the results obtained, what applications are foreseen for the developed material?

The authors answer: The interface material in this paper is mainly used for IC packaging and electronic heat dissipation which fill the micro-voids and uneven surface holes generated when the two materials join or contact, reduce the heat transfer contact thermal resistance and improve the device heat dissipation performance.

4) english must be revised both in syntax and grammar

The authors answer: We have revised and improved the grammar in the article.

5)when describing the figures, the number of the figure should be put in front of the letter: for example, Fig.1 (a,b) instead of Fig. (a,b)

The authors answer: We have made a targeted modification to this problem. (The red part in the paper)

6) some characterization techniques (for example the XRD technique) were not described in the manuscript while sufficient details were not provided for others

The authors answer: We have added a description of the relevant content in the article.  (The red part in 2.3 Detection and analysis)

7) how were the mechanical properties determined? please provide a description

The authors answer:  The interface material needs to apply a certain pressure when it is used to fill the gap between the heat source and the heat sink material. So the interface material needs to be compressible and the hardness of the material is closely related to the compressibility.

8) the XRD image inserted in figure 6f is absolutely not readable

The authors answer: We have made a targeted modification to this problem.

9) on the surface SEM images of the VACF/epoxy-0CNT interface material shown in Figure 4 (e,f), the scale bar which is absent must be inserted

The authors answer: We have made a targeted modification to this problem.

10)the most up-to-date references on the topic covered must be included in the manuscript; in this regard, I suggest consulting the following articles:

The authors answer: We have added relevant references. (The 31st reference)

Thank you very much for your attention and time. Look forward to hearing from you.

Yours sincerely,

Baoliu Li

24 January, 2024

Hubei Province Key Laboratory of Coal Conversion and New Carbon Materials,

Wuhan University of Science and Technology,

Wuhan 430081, China

Tel:86-27-68862591

mail: libaoliu@wust.edu.cn

Reviewer 3 Report

Comments and Suggestions for Authors

The subject/title of this manuscript, Preparation, microstructure and thermal properties of aligned mesophase pitch-based carbon fiber interface materials by an electrostatic flocking method, is very interesting.  However, in its current state it contains some limitations and more work is necessary!

 General comments:

  • The main problem with this manuscript is that parts of it read more like a technical report and lack the necessary quality for a research article.  For example, ALL 33 in the manuscript included references are ONLY cited in the introduction, NO references are cited in the results and discussion section; a more detailed discussion of the results is required, they should be compared to other in literature available data…”!
  • Explain abbreviations when first used.  For example, you state “SEM” in line 16 in the abstract, use the same abbreviation in the Figure 1 caption (line 95), before you explain this abbreviation in line 121 (“…scanning electron microscope (SEM).”).  Also, “XRD” is first introduced in line 76 but not explained.
  • Be consistent, use gaps between numbers and units.  For example, change “…24um…” to “24 μm…” (line 74); “…1.8mm.” to “…1.8 mm.” (line 75); etc.

 INTRODUCTION:  The authors cite 33 references in this section and lead the reader the purpose of their work.  

 EXPERIMENT:  Please include here information on the used plate distances, flocking voltages… (these parameters are only mentioned in the results and discussion section).  

 RESULTS AND DISCUSSION:  NO references are included in this section.  The authors must compare and discuss their results to in literature previously reported data!

 The nine included FIGURES are of good quality.

 The CONCLUSIONS can be improved.

 To summarise, this work has the potential for an interesting and timely study.  However, in its current state it contains limitations and more work is necessary.

Comments on the Quality of English Language

 Minor editing of English language required.

Author Response

Dear reviewer,

Thank you very much for your comments and professional advice. These opinions help to improve academic rigor of our article. Based on your suggestion and request, we have made corrected modifications on the revised manuscript. Meanwhile, the manuscript had be reviewed and edited by language of MDPI. We hope that our work can be improved again. Furthermore, we would like to show the details as follows:

  1. The main problem with this manuscript is that parts of it read more like a technical report and lack the necessary quality for a research article.  For example, ALL 33 in the manuscript included references are ONLY cited in the introduction, NO references are cited in the results and discussion section; a more detailed discussion of the results is required, they should be compared to other in literature available data…”!

The authors answer: We have added relevant references in the results and discussion section. We added reference data comparison in the main text section.

  1. Explain abbreviations when first used.  For example, you state “SEM” in line 16 in the abstract, use the same abbreviation in the Figure 1 caption (line 95), before you explain this abbreviation in line 121 (“…scanning electron microscope (SEM).”).  Also, “XRD” is first introduced in line 76 but not explained.

The authors answer: We have made a targeted modification to this problem and the test method of XRD was added. (The red part in Section 2.3. Detection and analysis)

  1. Be consistent, use gaps between numbers and units.  For example, change “…24um…” to “…24 μm…” (line 74); “…1.8mm.” to “…1.8 mm.” (line 75); etc.

The authors answer: We have made a targeted modification to this problem.  (The red part in the paper)

  1. EXPERIMENT:  Please include here information on the used plate distances, flocking voltages… (these parameters are only mentioned in the results and discussion section).  

The authors answer:  The plate spacing were 8 cm, 10 cm, 12 cm and the flocking voltage were 20 kV, 30 kV, 40 kV, respectively. The preparation process of the sample are added in the paper. (The red part in 2.1 Preparation of vertical array carbon fiber (VACF) )

  1. RESULTS AND DISCUSSION:  NO references are included in this section.  The authors must compare and discuss their results to in literature previously reported data!

The authors answer: We have added relevant references in the results and discussion section. We added reference data comparison in the results and discussion section.

  1. The CONCLUSIONS can be improved.

The author’s answer: We give an extended description of the conclusion section.

Thank you very much for your attention and time. Look forward to hearing from you.

Yours sincerely,

Baoliu Li

24 January, 2024

Hubei Province Key Laboratory of Coal Conversion and New Carbon Materials,

Wuhan University of Science and Technology,

Wuhan 430081, China

Tel:86-27-68862591

E-mail: libaoliu@wust.edu.cn

Round 2

Reviewer 2 Report

Comments and Suggestions for Authors

The authors have sufficiently addressed the issues raised by the reviewer. Therefore, the revised manuscript can be accepted in its current form.

Comments on the Quality of English Language

 Minor editing of English language required

Author Response

Dear reviewer,

Many thanks for having reviewed our manuscript and professional advice. We are grateful for the recognition of our efforts and appreciate your comments.

Yours sincerely,

Baoliu Li 

3 February, 2024

Hubei Province Key Laboratory of Coal Conversion and New Carbon Materials,

Wuhan University of Science and Technology,

Wuhan 430081, China

Tel:86-27-68862591

E-mail: libaoliu@wust.edu.cn

Reviewer 3 Report

Comments and Suggestions for Authors

Although the authors have improved their manuscript, they have not addressed the main problem with the manuscript, namely that ALL 34 in the manuscript included references are still ONLY cited in the introduction. The authors must compare and discuss their results to in literature previously reported data in the results and discussion section!

Comments on the Quality of English Language

Minor editing of English language required.

Author Response

Dear reviewer,

Thank you very much for your comments and professional advice. These opinions help to improve academic rigor of our article. Based on your suggestion and request, we have made corrected modifications on the revised manuscript. Meanwhile, the manuscript had be reviewed and edited by language of MDPI. We hope that our work can be improved again. Furthermore, we would like to show the details as follows:

Although the authors have improved their manuscript, they have not addressed the main problem with the manuscript, namely that ALL 34 in the manuscript included references are still ONLY cited in the introduction. The authors must compare and discuss their results to in literature previously reported data in the results and discussion section!

The author’s answer: We have added relevant references and data in the experiment, results and discussion section.(The red part in the paper)

Thank you very much for your attention and time. Look forward to hearing from you.

Yours sincerely,

Baoliu Li

3 February, 2024

Hubei Province Key Laboratory of Coal Conversion and New Carbon Materials,

Wuhan University of Science and Technology,

Wuhan 430081, China

Tel:86-27-68862591

E-mail: libaoliu@wust.edu.cn

Round 3

Reviewer 3 Report

Comments and Suggestions for Authors

The authors have further improved their manuscript, i.e. it can be accepted now.